# Paternal and maternal psychiatric history and risk of preterm and early term birth: A nationwide study using Swedish registers

Weiyao Yin[1,2]*, Jonas F. Ludvigsson[1,3], Ulrika Åden[4,5], Kari Risnes[6,7], Martina Persson[8,9,10], Abraham Reichenberg[11,12,13], Michael E. Silverman[11], Eero Kajantie[6,14,15], Sven Sandin[1,11,12]

1 Department of Medical Epidemiology and Biostatistics, Karolinska Institutet, Stockholm, Sweden, 2 Department of Obstetrics and Gynecology, West China Second University Hospital, Sichuan University, Chengdu, China, 3 Department of Pediatrics, Örebro University Hospital, Örebro, Sweden, 4 Department of Women's and Children's Health, Karolinska Institutet, Stockholm, Sweden, 5 Department of Biomedical and Clinical Sciences, Linköping University, Linköping, Sweden, 6 Department of Clinical and Molecular Medicine, NTNU, Trondheim, Norway, 7 Children's Clinic, St Olav University Hospital, Trondheim, Norway, 8 Department of Medicine, Clinical Epidemiological Unit, Karolinska Institutet, Stockholm, Sweden, 9 Department of Clinical Science and Education, Division of Pediatrics, Karolinska Institutet, Stockholm, Sweden, 10 Sachsska Childrens' and Youth Hospital, Stockholm, Sweden, 11 Department of Psychiatry, Icahn School of Medicine at Mount Sinai, New York, New York, United States of America, 12 Seaver Center for Autism Research and Treatment, Icahn School of Medicine at Mount Sinai, New York, New York, United States of America, 13 Department of Environmental Medicine and Public Health, Icahn School of Medicine at Mount Sinai, New York, New York, United States of America, 14 Population Health Unit, Finnish Institute for Health and Welfare, Helsinki and Oulu, Finland, 15 Clinical Medicine Research Unit, MRC Oulu, Oulu University Hospital and University of Oulu, Oulu, Finland

* weiyao.yin.2@ki.se

**Data Availability Statement:** Data cannot be shared publicly because of restrictions by law. Data are available from the upon re-quest to the Swedish registers for researchers who meet the criteria. (https://www.socialstyrelsen.se/en/

## Abstract

### Background

Women with psychiatric diagnoses are at increased risk of preterm birth (PTB), with potential life-long impact on offspring health. Less is known about the risk of PTB in offspring of fathers with psychiatric diagnoses, and for couples where both parents were diagnosed. In a nationwide birth cohort, we examined the association between psychiatric history in fathers, mothers, and both parents and gestational age.

### Methods and findings

We included all infants live-born to Nordic parents in 1997 to 2016 in Sweden. Psychiatric diagnoses were obtained from the National Patient Register. Data on gestational age were retrieved from the Medical Birth Register. Associations between parental psychiatric history and PTB were quantified by relative risk (RR) and two-sided 95% confidence intervals (CIs) from log-binomial regressions, by psychiatric disorders overall and by diagnostic categories. We extended the analysis beyond PTB by calculating risks over the whole distribution of gestational age, including "early term" (37 to 38 weeks).

Among the 1,488,920 infants born throughout the study period, 1,268,507 were born to parents without a psychiatric diagnosis, of whom 73,094 (5.8%) were born preterm. 4,597 of

statistics-and-data/registers/; Email contact: socialstyrelsen@socialstyrelsen.se).

**Funding:** The study was supported by grant from the Swedish Research Council (www.vr.se) 2021-0214 (grant for S.S.). The funders had no role in study design, data collection and analysis, decision to publish, or preparation of the manuscript.

**Competing interests:** I have read the journal's policy and the authors of this manuscript have the following competing interests: JL has coordinated a study on behalf of the Swedish IBD quality register (SWIBREG). That study received funding from Janssen corporation. JF has also received financial support from MSD (Merck Sharp & Dohme) developing a paper reviewing national healthcare registers in China. EK reported research grants by Academy of Finland, European Commission (733280 RECAP Research on European Children and Adults Born Preterm), Sigrid Juselius Foundation, Foundation for Pediatric Research, Signe and Ane Gyllenberg Foundation, Foundation for Diabetes Research, Foundation for Cardiovascular Research, Yrjö Jahnsson Foundation, Novo Nordisk Foundation. Other authors declared no competing interests. All authors confirm the independence of researchers from funders.

**Abbreviations:** BMI, body mass index; CI, confidence interval; ICD, International Classification of Diseases; IQR, interquartile range; MBR, Medical Birth Register; MGR, Multigeneration Register; NPR, National Patient Register; OCD, obsessive–compulsive disorder; OR, odds ratio; PTB, preterm birth; RR, relative risk; SEK, Swedish Krona.

73,500 (6.3%) infants were born preterm to fathers with a psychiatric diagnosis, 8,917 of 122,611 (7.3%) infants were born preterm to mothers with a pscyhiatric diagnosis, and 2,026 of 24,302 (8.3%) infants were born preterm to both parents with a pscyhiatric diagnosis. We observed a shift towards earlier gestational age in offspring of parents with psychiatric history. The risks of PTB associated with paternal and maternal psychiatric diagnoses were similar for different psychiatric disorders. The risks for PTB were estimated at RR 1.12 (95% CI [1.08, 1.15] $p < 0.001$) for paternal diagnoses, at RR 1.31 (95% CI [1.28, 1.34] $p < 0.001$) for maternal diagnoses, and at RR 1.52 (95% CI [1.46, 1.59] $p < 0.001$) when both parents were diagnosed with any psychiatric disorder, compared to when neither parent had a psychiatric diagnosis. Stress-related disorders were associated with the highest risks of PTB with corresponding RRs estimated at 1.23 (95% CI [1.16, 1.31] $p < 0.001$) for a psychiatry history in fathers, at 1.47 (95% CI [1.42, 1.53] $p < 0.001$) for mothers, and at 1.90 (95% CI [1.64, 2.20] $p < 0.001$) for both parents. The risks for early term were similar to PTB. Co-occurring diagnoses from different diagnostic categories increased risk; for fathers: RR 1.10 (95% CI [1.07, 1.13] $p < 0.001$), 1.15 (95% CI [1.09, 1.21] $p < 0.001$), and 1.33 (95% CI [1.23, 1.43] $p < 0.001$), for diagnoses in 1, 2, and $\geq$3 categories; for mothers: RR 1.25 (95% CI [1.22, 1.28] $p < 0.001$), 1.39 (95% CI [1.34, 1.44] $p < 0.001$) and 1.65 (95% CI [1.56, 1.74] $p < 0.001$). Despite the large sample size, statistical precision was limited in subgroups, mainly where both parents had specific psychiatric subtypes. Pathophysiology and genetics underlying different psychiatric diagnoses can be heterogeneous.

## Conclusions

Paternal and maternal psychiatric history were associated with a shift to earlier gestational age and increased risk of births before full term. The risk consistently increased when fathers had a positive history of different psychiatric disorders, increased further when mothers were diagnosed and was highest when both parents were diagnosed.

## Author summary

### Why was this study done?

- Women with psychiatric diagnoses are at increased risk of preterm birth (PTB). It is already known that PTB is associated with negative health consequences for the offspring.

- Less is known about the risk of PTB in offspring of fathers with psychiatric diagnoses and for couples where both parents had psychiatric diagnoses.

- Earlier studies have not thoroughly examined the full range of psychiatric disorders and gestational age.

**What did the researchers do and find?**

- In a cohort of 1.5 million births, we observed a shift towards earlier gestational age in offspring of parents with a history with psychiatric disorders, particularly for preterm and early term births.

- The risk of PTB consistently increased when fathers were diagnosed with different psychiatric disorders (relative risk (RR) = 1.12, 95% confidence interval (CI) [1.08, 1.15]), increased further when mothers were diagnosed (RR = 1.31, 95% CI [1.28, 1.34]), and was highest when both parents were diagnosed (RR = 1.52, 95% CI [1.46, 1.59]).

- For both the father and the mother, the risk increased in parents diagnosed with several different psychiatric disorders.

- The increased risk was present not only for children born preterm, but also for the larger group of offspring born at early term (37 to 38 weeks), who represent approximately 20% of all births.

**What do these findings mean?**

- These data suggest that the presence of psychiatric diagnoses in either one or both parents impacts gestational age at birth.

- Whether additional social and psychiatric support and prenatal care to families with a positive psychiatric history could mitigate against this warrants further investigation.

## Introduction

Higher rates of preterm births (<37 weeks gestation) resulting in increased mortality and adverse offspring outcomes [1–3] have been associated with maternal psychiatric history [4]. Although frequently attributed to environmental (antenatal distress, elevated neuro-endocrine activity, and homeostasis disruptions) [5] and genetic causes [1,6,7], critical knowledge gaps remain. First, the association between maternal psychiatric history and preterm delivery has not been extensively studied across a wide range of psychiatric disorders. One recent systematic review correlated mental illness before delivery with pregnancy complications including preterm delivery, with most evidence from depression and anxiety disorders [8]. Rather, most previous studies have generally focused on mood and anxiety disorders [8–14], with other psychiatric disorders receiving less attention [4,15–22]. Because these studies have typically relied on modest sample sizes and yielded mixed results, a comprehensive examination across different psychiatric disorders in a large population is required. Second, the association between paternal psychiatric history and preterm delivery has rarely been studied [23–25] and a rigorous study addressing the combined effect of psychiatric disorders in couples is notably absent. To understand the underlying mechanisms, it is essential to disentangle psychiatric history contributions in mothers from fathers. Third, evidence suggests that even moderately shorter gestation, as 1 or 2 weeks before full term, is associated with neonatal morbidity [26] and future health outcomes [2]. How the risk of preterm delivery in offspring of parents with psychiatric diagnoses extends to the entire range of gestational age is largely unknown,

particularly at early term (37 to 38 weeks). We are only aware of one study that has examined the risk of birth before full term for several psychiatric disorders in mothers [4]. No studies have examined changes in distribution of gestational age for paternal psychiatric disorders and shift of gestational age in preterm and early term births. Support for such an association will identify a large group at risk with unmet needs.

Using Swedish national registers, we examined the association between a wide range of clinically ascertained psychiatric diagnoses in fathers, mothers, and both parents and gestational age in offspring, compared to offspring born to parents without psychiatric diagnoses.

## Methods

### Ethics statement

The study was approved by the national Swedish ethics review board, Sweden (2017/1875-31/1; 2018/1864-32; 2019–06314).

### Data sources and study population

We included all infants born alive in Sweden from 1997 to 2016, to parents born in Sweden, Finland, Norway, Denmark, and Iceland, and registered in the Swedish Medical Birth Register (MBR) [27]. Fathers were identified by linkage to the Swedish Multigeneration Register (MGR). Data on parental education (completed school years) and income were extracted from the Swedish government's database for health insurance and labor market studies (LISA database) [28]. Psychiatric diagnoses from specialist care were extracted from the Swedish National Patient Register (NPR) according to the International Classification of Diseases (ICD) (inpatient diagnoses since 1973 and nationwide coverage of outpatient diagnoses since approximately 2001). The data quality of the NPR has been verified [29] and validated for diagnoses of psychiatric conditions [30]. Data on vital status and emigration was derived from the Total Population Register. Individual-level data from different registers were linked using the Swedish unique personal identity number of each citizen. The data were made available from the European Union's Horizon 2020 research and innovation program "RECAP preterm" (Research on European Children and Adults born preterm, www.recap-preterm.eu).

### Psychiatric diagnoses and preterm birth

We defined a psychiatric history in parents as any first psychiatric diagnosis (Table A in S1 Appendix) registered in the NPR any time before delivery. We divided psychiatric disorders in 6 diagnostic categories: psychoactive substance use, schizophrenia and other non-mood psychotic disorder, mood disorders, neurotic/behavioral disorders, neurodevelopmental disorders (including emotional and behavioral disorders of childhood origin and intellectual disability), and other psychiatric disorder (not otherwise specified). Several subtypes under the diagnostic categories were further examined, including depression, bipolar, anxiety, obsessive–compulsive disorder (OCD), stress-related disorders, somatoform, and eating disorders. Individuals with several types of psychiatric diagnoses can contribute to different diagnostic categories.

Gestational age in days was collected from the MBR. Since 1990, early second-trimester ultrasound examination is routinely offered to determine gestational age, and more than 95% of women accepted this offer; otherwise, the date of last menstrual period is used [31]. We categorized gestational age into preterm (<37 weeks) and term (≥37 weeks) birth and then further into very preterm (<32 weeks), moderate to late preterm (32 to 36 weeks), early term (37 to 38 weeks), and full term (≥39 weeks) [26]. We classified deliveries as spontaneous and non-

spontaneous, i.e., spontaneous initiation of labor, or through induction of labor or cesarean section, according to the record of onset of labor in the MBR.

## Other covariates

Birth year, sex, maternal age, maternal cigarette smoking during pregnancy (yes/no), maternal body mass index (BMI) at the first prenatal visit ($<18.5$ kg/m$^2$, 18.5 to 24.9 kg/m$^2$, 25 to 29.9 kg/m$^2$, $\geq 30$ kg/m$^2$) and onset of delivery (spontaneous versus non-spontaneous delivery) were obtained from the MBR. Paternal age and parental yearly income (Swedish Krona, SEK) and level of education ($<9$ years primary education, 9 years primary education, 1 to 2 years secondary education, 3 years secondary education, 1 to 2 years postgraduate education, $\geq 3$ years postgraduate education, PhD education) were defined at delivery.

## Statistical analyses

To address potential changes in the full gestational age distribution, without dichotomizing the data, we showed the shape of the gestational age distribution for the different exposure groups by calculating probability density functions. A density curve provides a visualization of what percentage of observations in the gestational age distribution fall between different values (here: gestational weeks). Next, we fitted quantile regression models adjusting for offspring's birth year (categorized in 5-year intervals) to examine the effects of exposures at different quantiles. Whereas a linear regression analyses differences in the mean between exposures, quantile regression can compare exposures at an arbitrary cutoff of a data distribution, e.g., at the 1st, 5th, or 10th percentiles, and study effects on the most sensitive lower tail of the gestational age distribution. Thus, for the analysis of gestational age, quantile regression can estimate at which gestational age (week) the 1%, 5%, or 10% earliest born children were delivered and how this age differs between exposures.

We quantified the association between parental psychiatric history and preterm birth by relative risks (RRs) and associated two-sided 95% confidence intervals (CIs) from log-binomial regression models. In primary analyses, we examined any psychiatric diagnosis present in both parents, only in mothers and only in fathers separately, adjusted for offspring birth year as natural cubic splines with 5 degrees of freedom (Model 1) [32]. We then examined the risk by 6 psychiatric diagnostic categories and by subtypes within each categories. Forest plots were created to depict the trends of RRs by different psychiatric diagnoses in the parents. We repeated the analyses in finer categories of gestational age, including very preterm, moderate/late preterm, early term, and full term, with risks quantified by odds ratio (OR) from multinomial logistic regression. We examined the preterm risk in relation to the number of co-occurring psychiatric diagnoses in different diagnostic categories, by diagnoses in 1, 2, or $\geq 3$ categories, in mothers and fathers separately.

All statistical tests were performed on the two-sided 5% level of significance. To address potential correlation between siblings, robust standard errors were applied [33]. We did not adjust the $p$-values for multiplicity of statistical tests. However, the primary research question may be addressed by using only 3 sequential statistical tests (psychiatric history in both parents, only in mothers and only in fathers, versus in neither parent).

## Supplementary analyses

(1) We fitted a sequence of models adjusting for paternal and maternal age, education, and income as well as maternal BMI in early pregnancy and smoking during pregnancy [1,34]. Since these factors may influence the associations, not only as confounders between psychiatric diagnosis and gestational age, but also as modifiers and/or mediators, adjustment in

regression analysis may introduce unknown biases; therefore, we did not include them in the primary analysis (Model 1; only including birth year as covariate). Still, to explore if these covariates provide any major influences on the association without resolving their exact potential roles as confounders, mediators, or modifiers, we adjusted for them in supplementary models. In Model 2, we adjusted for covariates related with both maternal and paternal psychiatric diagnosis, including maternal and paternal age (as natural cubic splines), level of education and income (modeled by ranks as natural cubic splines) at delivery. Next, in Model 3, we further adjusted for maternal-specific covariates, including maternal smoking and BMI. (2) We performed pairwise comparisons between psychiatric exposure groups, comparing risk of preterm delivery associated with psychiatric history in both parents to only in fathers, only in mothers to only in fathers, and in both parents to only in mothers. (3) We stratified analyses by spontaneous delivery and non-spontaneous. (4) We estimated the preterm risk separately by offspring sex and by including an interaction term between psychiatric history and sex. (5) We restricted the analyses to singletons. (6) To examine the specificity of timing of the diagnosis, we restricted the analysis to parental psychiatric disorders first diagnosed prior to 2 years before conception. (7) To determine the extent to which the results were driven by individuals with provisional diagnoses, i.e., only 1 single diagnosis in an individual or diagnoses given over a short period of time, we repeated the main analysis but excluded parental psychiatric diagnoses recorded only once in the patient register or some diagnoses within 30 days. (8) We restricted the population to births from 2005 onwards.

Statistical analyses were performed using SAS software version 9.4 (SAS Institute, Cary, North Carolina, United States of America). This study was reported as per the Strengthening the Reporting of Observational Studies in Epidemiology (STROBE) guideline.

## Results

Of 1,499,701 infants, we excluded 32 with implausible death/emigration date. Only 10,749 (0.7%) had missing data on covariates (gestational age, sex, parental income and education) and were excluded from the analyses. Thus, our analytic cohort comprised 1,488,920 infants (Fig A in S1 Appendix). A total of 1,268,507 infants were born to parents without a psychiatric diagnosis, of whom 73,094 (5.8%) were born preterm. A total of 4,597 of 73,500 (6.3%) infants were born preterm to fathers with a psychiatric diagnosis, 8,917 of 122,611 (7.3%) infants were born preterm to mothers with a psychiatric diagnosis, and 2,026 of 24,302 (8.3%) infants were born preterm to both parents with a psychiatric diagnosis. The median lag years between the first observed diagnosis and the date of delivery was 7 years (interquartile range (IQR): 4 to 13 years) in fathers and 6 years (IQR: 3 to 11 years) in mothers; the corresponding years for the last observed diagnosis was 2 years (IQR: 0.6 to 6 years) in fathers and 2 years (IQR: 0.6 to 5 years) in mothers. The cumulative distribution of years between birth of the child and last previous diagnosis for mothers and fathers separately is shown in Fig C in S1 Appendix. Based on summary statistics, offspring of parents with psychiatric history tended to be born in recent years and to younger parents with lower level of education and income. Their mothers were more likely to be underweight or obese, to have smoked during pregnancy, and to have spontaneous labor (Table 1).

### Parental psychiatric history before delivery, gestational age, and preterm birth

Overall, compared to offspring of neither parent with a psychiatric diagnosis, the gestational age distribution was shifted towards earlier gestational age for offspring of parents with psychiatric history, particularly for a psychiatric history in mothers and in both parents and for the

**Table 1. Cohort characteristics.**

| | Psychiatric history in parents | | | |
|---|---|---|---|---|
| Characteristics | Neither parents | Fathers only | Mothers only | Both parents |
| | Number of infants (%) | Number of infants (%) | Number of infants (%) | Number of infants (%) |
| Total number | 1,268,507 | 73,500 | 122,611 | 24,302 |
| Preterm birth (<37 week) | 73,094 (5.76) | 4,597 (6.25) | 8,917 (7.27) | 2,026 (8.34) |
| Gestational age | | | | |
| <32 week | 10,510 (0.83) | 725 (0.99) | 1,366 (1.11) | 292 (1.20) |
| 32–36 week | 62,584 (4.93) | 3,872 (5.27) | 7,551 (6.16) | 1,734 (7.14) |
| 37–38 | 230,854 (18.20) | 13,604 (18.51) | 27,656 (22.56) | 5,625 (23.15) |
| ≥39 week | 964,559 (76.04) | 55,299 (75.24) | 86,038 (70.17) | 16,651 (68.52) |
| Birth year | | | | |
| 1997–2001 | 324,128 (25.55) | 9,200 (12.52) | 10,859 (8.86) | 1,287 (5.30) |
| 2002–2006 | 342,665 (27.01) | 13,429 (18.27) | 20,192 (16.47) | 2,638 (10.86) |
| 2007–2011 | 322,125 (25.39) | 21,377 (29.08) | 38,137 (31.10) | 7,061 (29.06) |
| 2012–2016 | 279,589 (22.04) | 29,494 (40.13) | 53,423 (43.57) | 13,316 (54.79) |
| Offspring sex (male, %) | 653,129 (51.49) | 37,723 (51.32) | 63,104 (51.47) | 12,526 (51.54) |
| Maternal age at delivery (years; median, IQR) | 31 (28–34) | 30 (26–34) | 30 (26–34) | 29 (25–33) |
| Paternal age at delivery (years; median, IQR) | 33 (30–37) | 33 (28–37) | 32 (28–37) | 32 (27–37) |
| Maternal yearly income (SEK; median, IQR) | 143,262 (109,037–300,429) | 151,339 (112,681–281,916) | 146,363 (107,915–313,642) | 140,273 (102,177–254,303) |
| Paternal yearly income (SEK; median, IQR) | 227,213 (166,302–300,429) | 209,453 (138,370–281,916) | 244,473 (176,760–313,642) | 178,378 (111,472–254,303) |
| Maternal education | | | | |
| <9 years primary school | 1,000 (0.08) | 271 (0.37) | 675 (0.55) | 432 (1.78) |
| 9 years primary school | 78,209 (6.17) | 10,255 (13.95) | 19,767 (16.12) | 7,508 (30.89) |
| 1–2 years secondary school | 183,903 (14.50) | 10,138 (13.79) | 16,790 (13.69) | 4,166 (17.14) |
| 3 years secondary school | 371,272 (29.27) | 25,829 (35.14) | 37,296 (30.42) | 7,044 (28.99) |
| 1–2 years postgraduate education | 181,675 (14.32) | 8,173 (11.12) | 14,469 (11.80) | 1,960 (8.07) |
| ≥3 years postgraduate education | 442,649 (34.90) | 18,516 (25.19) | 33,080 (26.98) | 3,158 (12.99) |
| PhD | 9,799 (0.77) | 318 (0.43) | 534 (0.44) | 34 (0.14) |
| Paternal education | | | | |
| <9 years primary school | 2,678 (0.21) | 942 (1.28) | 488 (0.40) | 636 (2.62) |
| 9 years primary school | 106,526 (8.40) | 16,148 (21.97) | 14,593 (11.90) | 7,641 (31.44) |
| 1–2 years secondary school | 291,240 (22.96) | 17,065 (23.22) | 22,691 (18.51) | 5,666 (23.31) |
| 3 years secondary school | 363,005 (28.62) | 21,353 (29.05) | 45,366 (37.00) | 6,793 (27.95) |
| 1–2 years postgraduate education | 191,827 (15.12) | 7,082 (9.64) | 15,013 (12.24) | 1,679 (6.91) |
| ≥3 years postgraduate education | 297,414 (23.45) | 10,480 (14.26) | 23,411 (19.09) | 1,814 (7.46) |
| PhD | 15,817 (1.25) | 430 (0.59) | 1,049 (0.86) | 73 (0.30) |
| Maternal smoking during pregnancy | | | | |
| Yes | 64,194 (5.06) | 8,574 (11.67) | 13,207 (10.77) | 5,876 (24.18) |
| No | 1,146,709 (90.40) | 62,079 (84.46) | 104,437 (85.18) | 17,480 (71.93) |
| Unknown | 57,604 (4.54) | 2,847 (3.87) | 4,967 (4.05) | 946 (3.89) |
| Maternal body mass index (kg/m$^2$) at the first prenatal visit | | | | |
| <18.5 | 21,533 (1.70) | 1,660 (2.26) | 3,103 (2.53) | 797 (3.28) |
| 18.5–24.9 | 712,555 (56.17) | 37,773 (51.39) | 64,963 (52.98) | 11,851 (48.77) |
| 25–29.9 | 282,896 (22.30) | 17,169 (23.36) | 27,905 (22.76) | 5,659 (23.29) |
| ≥30 | 123,747 (9.76) | 10,476 (14.25) | 15,761 (12.85) | 3,867 (15.91) |
| Unknown | 127,776 (10.07) | 6,422 (8.74) | 10,879 (8.87) | 2,128 (8.76) |
| Spontaneous delivery (yes, %) | 1,002,232 (79.01) | 56,654 (77.08) | 86,961 (70.92) | 17,107 (70.39) |

(*Continued*)

**Table 1.** (Continued)

| Characteristics | Psychiatric history in parents | | | |
| --- | --- | --- | --- | --- |
| | Neither parents | Fathers only | Mothers only | Both parents |
| | Number of infants (%) | Number of infants (%) | Number of infants (%) | Number of infants (%) |
| Multiple births (yes, %) | 38,027 (3.00) | 2,138 (2.91) | 3,687 (3.01) | 692 (2.85) |

BMI, body mass index; IQR, interquartile range; SEK, Swedish Krona.

lower quantiles of gestational age, e.g., quantiles lower than 10% tile, approximately 38 weeks of gestational age (Fig 1 and Fig B in S1 Appendix).

Adjusted for birth year, for any psychiatric diagnosis, the RR for preterm birth was estimated at RR = 1.12 (95% CI [1.08, 1.15] $p < 0.001$) for fathers, at RR = 1.31 (95% CI [1.28, 1.34] $p < 0.001$) for mothers, and at RR = 1.52 (95% CI [1.46, 1.59] $p < 0.001$) for psychiatric history in both parents, compared to when neither parent had a psychiatric diagnosis. Additionally, adjusting for maternal and paternal age, income, and education, the RR was similar

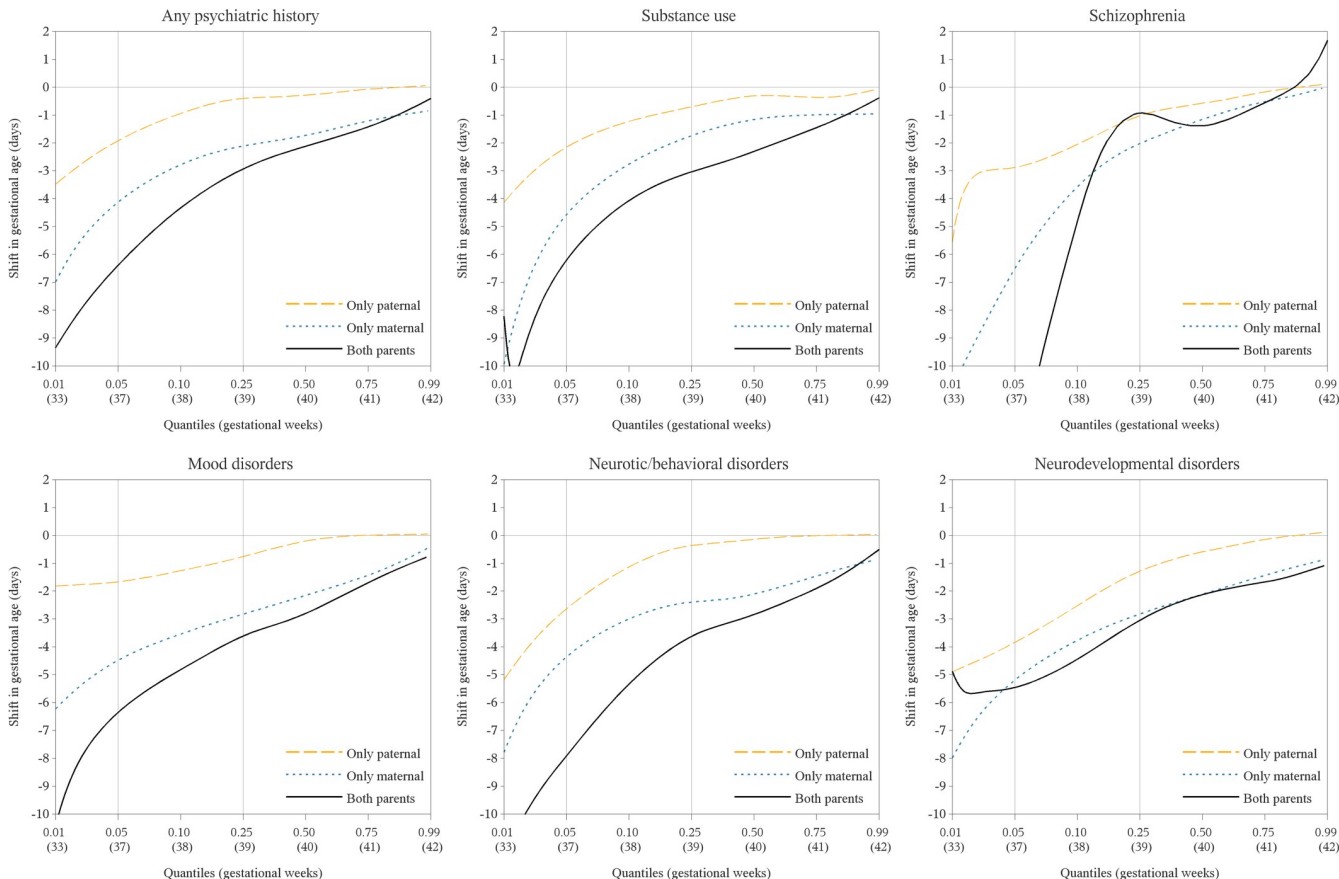

**Fig 1. Differences in gestational age (days) for births to mothers, fathers, and both parents with psychiatric diagnosis compared to that of neither parent with psychiatric diagnosis (reference group) at each different percentile.** Percentile: A division of a data distribution into 100 parts where each part contains 1% of the data. For example, the lowest 5% of the data are found below the fifth percentile. For each psychiatric diagnostic category, the quantile regression models examined the effects of exposures at each different percentile of GA, comparing GA for births of mothers, fathers, and both parents with psychiatric diagnosis, to that of neither parent with psychiatric diagnosis, adjusted for offspring's birth year in 5-year interval. The quantile regression estimates the shift of GA at the lower tails of the GA distribution, i.e., the most sensitive periods of preterm and early term births. To facilitate interpretation, the GA (approximately weeks) for the reference group is printed underneath each percentile. GA, gestational age.

for maternal associations 1.27 (95% CI [1.24, 1.29] $p < 0.001$), but noticeably reduced for paternal associations RR = 1.04 (95% CI [1.01, 1.07] $p = 0.006$) (Table 2). Adjusting further for maternal BMI and maternal smoking did not further alter the results (Table C in S1 Appendix).

The risk patterns remained consistent across different psychiatric disorders (Fig 2). Stress-related disorders were associated with the highest risks of preterm birth with corresponding RRs estimated at 1.23 (95% CI [1.16, 1.31] $p < 0.001$) for a psychiatry history in fathers, at 1.47 (95% CI [1.42, 1.53] $p < 0.001$) for mothers, and at 1.90 (95% CI [1.64, 2.20] $p < 0.001$) for both parents (Table 2, Model 1). The patterns of risk for preterm birth were also seen in infants born very preterm, moderate/late preterm, and early term (Fig 3A and Table D in S1 Appendix). Co-occurring diagnoses from different diagnostic categories increased risk; for fathers: RR = 1.10 (95% CI [1.07, 1.13] $p < 0.001$), 1.15 (95% CI [1.09, 1.21] $p < 0.001$), and 1.33 (95% CI [1.23, 1.43] $p < 0.001$), for diagnoses in 1, 2, and $\geq 3$ categories; for mothers: RR = 1.25 (95% CI [1.22, 1.28] $p < 0.001$), 1.39 (95% CI [1.34, 1.44] $p < 0.001$), and 1.65 (95% CI [1.56, 1.74] $p < 0.001$) (Fig 3B and Table E in S1 Appendix).

## Supplementary analysis

In the pairwise comparisons between exposure groups, when compared to offspring of fathers only with psychiatric history, the preterm risk was statistically significantly higher for offspring where mothers only had a psychiatric history; RR = 1.17 (95% CI [1.13, 1.21] $p < 0.001$), and highest when both parents had psychiatric history; RR = 1.36 (95% CI [1.29, 1.43] $p < 0.001$) (Table F in S1 Appendix). Compared to offspring of mothers with psychiatric history, the risk was statistically significantly higher for offspring where both parents had a psychiatric history; RR = 1.16 (95% CI [1.11, 1.22] $p < 0.001$). The RRs were higher for spontaneous deliveries (for psychiatric history in both parents: RR = 1.57 (95% CI [1.48, 1.66] $p < 0.001$), compared to non-spontaneous deliveries: RR = 1.28 (95% CI [1.20, 1.37] $p < 0.001$), supported by non-overlapped confidence intervals (Table G in S1 Appendix). The interaction between offspring sex and parental psychiatric history was not statistically significant ($p$-value = 0.42; Table H in S1 Appendix). When restricting to singleton births (Table I in S1 Appendix), to psychiatric diagnoses first being diagnosed prior to 2 years before conception (Table J in S1 Appendix), to diagnoses occurred at least twice lasting more than 30 days (Table K in S1 Appendix), and to births from 2005 onwards (Table L in S1 Appendix), the estimated preterm risks were essentially similar to that in the main analysis.

## Discussion

Gestational age is an established predictor for morbidity and mortality across life span. In this nationwide birth cohort study spanning over 20 years, including 1.5 million births, we observed a shift towards earlier gestational age in offspring of parents with a history of psychiatric diagnosis, particularly at lower quantiles of gestational age corresponding to preterm and early term births. Consequently, the risks of preterm and early term birth were increased, with increased risk when only fathers had a psychiatric history, even higher risks when only mothers had a psychiatric history and highest for psychiatric history in both parents, compared to neither parent with psychiatric history. One dose-dependent pattern was seen with increasing number of co-occurring psychiatric disorders from different diagnostic categories in fathers and mothers. After adjustment for parental age and socioeconomic status and maternal BMI and smoking, the risk was largely similar for maternal associations but was noticeably reduced for paternal associations, although it remained statistically significant. The preterm risk was higher among spontaneous deliveries, compared to non-spontaneous deliveries. There were no sex-specific differences in risk.

**Table 2. Parental psychiatric diagnosis before delivery and RR of preterm birth.**

| Analysis group | Preterm births (%) | Number of subjects | Model 1 RR (95% CI) | Model 1 P value | Model 1 RR (95% CI) | Model 2 P value |
|---|---|---|---|---|---|---|
| **Any psychiatric diagnosis** | | | | | | |
| Neither parent | 73,094 (5.76) | 1,268,507 | Reference | | Reference | |
| Fathers only | 4,597 (6.25) | 73,500 | 1.12 (1.08, 1.15) | <0.001 | 1.04 (1.01, 1.07) | 0.006 |
| Mothers only | 8,917 (7.27) | 122,611 | 1.31 (1.28, 1.34) | <0.001 | 1.27 (1.24, 1.29) | <0.001 |
| Both parents | 2,026 (8.34) | 24,302 | 1.52 (1.46, 1.59) | <0.001 | 1.36 (1.30, 1.42) | <0.001 |
| **Substance use** | | | | | | |
| Neither parent | 83,578 (5.89) | 1,418,370 | Reference | | Reference | |
| Fathers only | 2,393 (6.67) | 35,867 | 1.15 (1.11, 1.20) | <0.001 | 1.05 (1.00, 1.09) | 0.031 |
| Mothers only | 2,204 (7.53) | 29,252 | 1.31 (1.26, 1.36) | <0.001 | 1.22 (1.17, 1.27) | <0.001 |
| Both parents | 459 (8.45) | 5,431 | 1.47 (1.35, 1.61) | <0.001 | 1.26 (1.16, 1.38) | <0.001 |
| **Schizophrenia and other non-mood psychotic disorder** | | | | | | |
| Neither parent | 88,126 (5.95) | 1,482,315 | Reference | | Reference | |
| Fathers only | 233 (7.20) | 3,236 | 1.22 (1.08, 1.38) | 0.002 | 1.11 (0.98, 1.26) | 0.095 |
| Mothers only | 266 (8.12) | 3,274 | 1.37 (1.22, 1.54) | <0.001 | 1.30 (1.16, 1.46) | <0.001 |
| Both parents | 9 (9.47) | 95 | 1.60 (0.86, 2.98) | 0.137 | 1.39 (0.74, 2.60) | 0.309 |
| **Mood disorders** | | | | | | |
| Neither parent | 83,138 (5.87) | 1,415,358 | Reference | | Reference | |
| Fathers only | 1,363 (6.57) | 20,750 | 1.15 (1.09, 1.21) | <0.001 | 1.06 (1.01, 1.12) | 0.027 |
| Mothers only | 3,814 (7.76) | 49,152 | 1.37 (1.33, 1.41) | <0.001 | 1.32 (1.28, 1.36) | <0.001 |
| Both parents | 319 (8.72) | 3,66 | 1.56 (1.40, 1.73) | <0.001 | 1.38 (1.24, 1.53) | <0.001 |
| **Depression** | | | | | | |
| Neither parent | 83,538 (5.88) | 1,420,812 | Reference | | Reference | |
| Fathers only | 1,254 (6.60) | 19,001 | 1.16 (1.10, 1.22) | <0.001 | 1.06 (1.01, 1.12) | 0.028 |
| Mothers only | 3,565 (7.75) | 45,975 | 1.37 (1.32, 1.41) | <0.001 | 1.32 (1.28, 1.36) | <0.001 |
| Both parents | 277 (8.84) | 3,132 | 1.58 (1.41, 1.76) | <0.001 | 1.40 (1.25, 1.56) | <0.001 |
| **Bipolar** | | | | | | |
| Neither parent | 87,941 (5.94) | 1,480,341 | Reference | | Reference | |
| Fathers only | 196 (7.02) | 2,791 | 1.22 (1.06, 1.39) | 0.005 | 1.12 (0.98, 1.28) | 0.103 |
| Mothers only | 487 (8.55) | 5,699 | 1.49 (1.37, 1.62) | <0.001 | 1.42 (1.30, 1.55) | <0.001 |
| Both parents | 10 (11.24) | 89 | 1.98 (1.11, 3.56) | 0.021 | 1.73 (0.97, 3.10) | 0.065 |
| **Neurotic/behavioral disorders** | | | | | | |
| Neither parent | 78,269 (5.81) | 1,347,520 | Reference | | Reference | |
| Fathers only | 2,660 (6.48) | 41,038 | 1.15 (1.11, 1.19) | <0.001 | 1.07 (1.03, 1.12) | <0.001 |
| Mothers only | 6,881 (7.53) | 91,354 | 1.34 (1.31, 1.37) | <0.001 | 1.30 (1.27, 1.33) | <0.001 |
| Both parents | 824 (9.15) | 9,008 | 1.65 (1.55, 1.76) | <0.001 | 1.48 (1.38, 1.58) | <0.001 |
| **Anxiety** | | | | | | |
| Neither parent | 84,047 (5.89) | 1,426,539 | Reference | | Reference | |
| Fathers only | 1,172 (6.49) | 18,071 | 1.14 (1.08, 1.21) | <0.001 | 1.05 (1.00, 1.12) | 0.064 |
| Mothers only | 3,154 (7.61) | 41,422 | 1.35 (1.30, 1.39) | <0.001 | 1.30 (1.25, 1.34) | <0.001 |
| Both parents | 261 (9.04) | 2,888 | 1.62 (1.44, 1.82) | <0.001 | 1.42 (1.27, 1.60) | <0.001 |
| **OCD** | | | | | | |
| Neither parent | 88,130 (5.95) | 1,482,236 | Reference | | Reference | |
| Fathers only | 129 (6.68) | 1,932 | 1.16 (0.98, 1.37) | 0.081 | 1.11 (0.94, 1.31) | 0.237 |
| Mothers only | 374 (7.92) | 4,723 | 1.38 (1.25, 1.52) | <0.001 | 1.35 (1.23, 1.49) | <0.001 |
| Both parents | 1 (3.45) | 29 | 0.60 (0.09, 4.13) | 0.607 | 0.59 (0.09, 4.08) | 0.593 |
| **Stress-related** | | | | | | |
| Neither parent | 84,856 (5.88) | 1,442,247 | Reference | | Reference | |

*(Continued)*

**Table 2.** (Continued)

| Analysis group | Preterm births (%) | Number of subjects | Model 1 RR (95% CI) | Model 1 P value | Model 1 RR (95% CI) | Model 2 P value |
|---|---|---|---|---|---|---|
| Fathers only | 1,015 (7.07) | 14,349 | 1.23 (1.16, 1.31) | <0.001 | 1.13 (1.06, 1.20) | <0.001 |
| Mothers only | 2,605 (8.44) | 30,858 | 1.47 (1.42, 1.53) | <0.001 | 1.39 (1.34, 1.44) | <0.001 |
| Both parents | 158 (10.78) | 1,466 | 1.90 (1.64, 2.20) | <0.001 | 1.65 (1.42, 1.91) | <0.001 |
| **Somatoform** | | | | | | |
| Neither parent | 87,337 (5.94) | 1,470,942 | Reference | | Reference | |
| Fathers only | 433 (6.39) | 6,780 | 1.09 (0.99, 1.19) | 0.072 | 1.05 (0.96, 1.16) | 0.250 |
| Mothers only | 860 (7.73) | 11,125 | 1.31 (1.23, 1.40) | <0.001 | 1.27 (1.19, 1.35) | <0.001 |
| Both parents | 4 (5.48) | 73 | 0.94 (0.36, 2.45) | 0.906 | 0.89 (0.34, 2.32) | 0.806 |
| **Eating disorders** | | | | | | |
| Neither parent | 87,360 (5.94) | 1,470,936 | Reference | | Reference | |
| Fathers only | 134 (6.43) | 2,083 | 1.11 (0.94, 1.31) | 0.199 | 1.07 (0.91, 1.26) | 0.427 |
| Mothers only | 1,139 (7.18) | 15,862 | 1.24 (1.17, 1.31) | <0.001 | 1.22 (1.16, 1.30) | <0.001 |
| Both parents | 1 (2.56) | 39 | 0.45 (0.06, 3.10) | 0.416 | 0.45 (0.06, 3.08) | 0.413 |
| **Neurodevelopmental disorders** | | | | | | |
| Neither parent | 86,337 (5.92) | 1,459,349 | Reference | | Reference | |
| Fathers only | 974 (7.27) | 13,394 | 1.27 (1.20, 1.35) | <0.001 | 1.11 (1.05, 1.19) | <0.001 |
| Mothers only | 1,181 (8.12) | 14,537 | 1.42 (1.34, 1.50) | <0.001 | 1.29 (1.22, 1.36) | <0.001 |
| Both parents | 142 (8.66) | 1,640 | 1.54 (1.32, 1.81) | <0.001 | 1.26 (1.08, 1.48) | 0.004 |
| **Other** | | | | | | |
| Neither parent | 88,374 (5.95) | 1,486,195 | Reference | | Reference | |
| Fathers only | 76 (7.99) | 951 | 1.39 (1.12, 1.72) | 0.003 | 1.24 (1.00, 1.54) | 0.049 |
| Mothers only | 180 (10.33) | 1,742 | 1.79 (1.56, 2.06) | <0.001 | 1.64 (1.43, 1.89) | <0.001 |
| Both parents | 4 (12.50) | 32 | 2.19 (0.87, 5.48) | 0.095 | 1.88 (0.74, 4.75) | 0.184 |

RRs with 95% CIs were calculated using log-binomial regression models with robust standard errors. Model 1: Adjusted for birth year as cubic natural splines with 5 knots; Model 2: Additionally adjusted for maternal and paternal education (<9 years of primary education, 9 years of primary education, 1–2 years of secondary school education, 3 years of secondary school education, 1–2 years of postgraduate education, ≥3 years of postgraduate education, PhD education), yearly income (SEK; modeled by ranks as natural cubic splines with 5 degrees of freedom), and age (years; as natural cubic splines with 5 degrees of freedom), all defined at delivery.

CI, confidence interval; OCD, obsessive–compulsive disorder; RR, relative risk.

Strengths of our study include the large population-based cohort design, with prospective ascertainment of psychiatric diagnoses assigned by specialists and gestational age determined using ultrasound. Selection bias was minimized by utilizing data from a publicly financed health care system with universal access and close to complete follow-up. Compared to earlier studies, our sample size and detailed database enabled us to examine risks associated with a wide range of psychiatric disorders in one population and to examine, to the best of our knowledge, the additive effect of maternal and paternal exposure for the first time.

Our study also have several limitation. Despite the large sample size, statistical precision was limited in subgroups, mainly where both parents had specific psychiatric subtypes. Although we examined the risk by diagnostic categories and subtypes of psychiatric disorders, pathophysiology and genetics underlying different categories can still be very heterogeneous. We do not have data on psychiatric diagnoses set in primary care, which excludes subclinical conditions. Possibly these individuals can introduce measurement errors that could result in, if anything, underestimating the actual associations. We do not have information on disease severity and activity during pregnancy. However, we examined the co-occurrence of several psychiatric diagnoses as an indicator of disease load. Future studies will have to address the potential impact of psychiatric medications or other types of treatment. Finally, we cannot rule

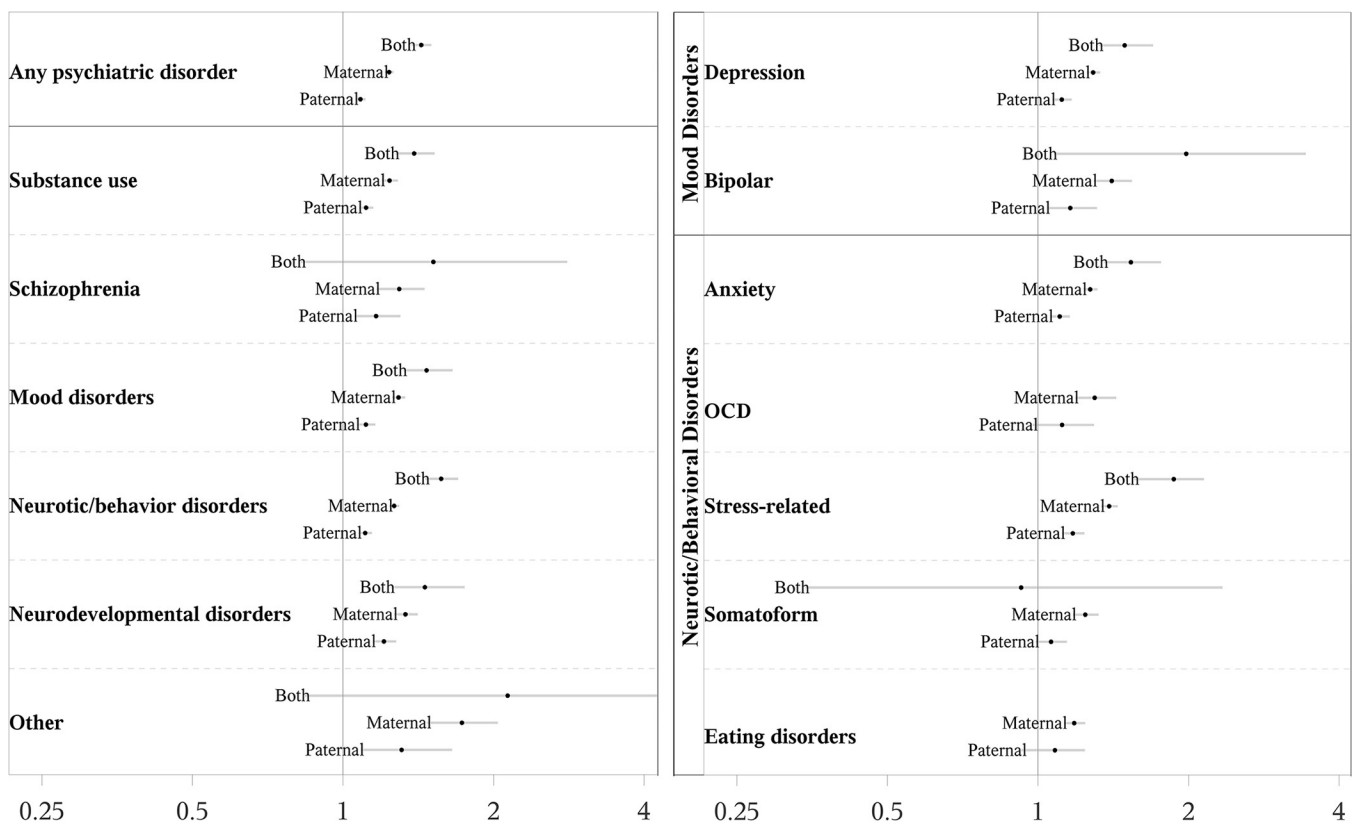

**Fig 2. Parental psychiatric history before delivery and RR of preterm birth.** RRs with 95% CIs were calculated using log-binomial regression models, adjusted for birth year by cubic natural splines with 5 knots. The dots in the figure represent RRs and lines represent 95% CIs. For psychiatric category "other," upper 95% CI = 5.48. CI, confidence interval; P.hist, psychiatric history; RR, relative risk.

out residual confounding, e.g., due to maternal infection or other unknown factors, as well as bias due to fetal death, miscarriage, or abortion at extremely early gestational age range.

This is, as far as we know, the first large population-based study to comprehensively examine the full gestational range by psychiatric diagnoses for both the parents. Importantly, the increased risk was present not only for the 5% of the offspring born preterm, but also for the larger group of offspring born at early term—in Sweden approximately 20% of all births, and remained consistent across different psychiatric disorders. There is a growing literature supporting the importance of delivery in the optimal biological window around week 40. For example, morbidity and mortality are elevated not only for infants born preterm, but also for infants born "early term" [2,3], which opens a larger pathway of risk associated with gestational age, not only for a small subset of the population. Only 1 study examined the risk of birth before full term. It included only mothers with psychiatric disorders but reported an increased risk of births before 39 weeks, consistent with our finding of an increased risk of births at early term between 37 and 38 weeks [4]. Two earlier studies on prenatal paternal depression and schizophrenia suggested paternal depression as a risk factor for preterm birth [23], but not for paternal schizophrenia [24]. In our study, we observed increased risk of preterm birth in cases of any paternal psychiatric diagnosis and by a wide range of psychiatric diagnostic categories. Increased risk of preterm delivery in mothers with a psychiatric history has been reported in several studies [4,20,35–39]. The present study supported earlier findings and extended previous data by examining risks in 13 diagnostic categories/subtypes. Our results are in line with studies reporting higher preterm risks with maternal mood and anxiety disorders [8–11,14],

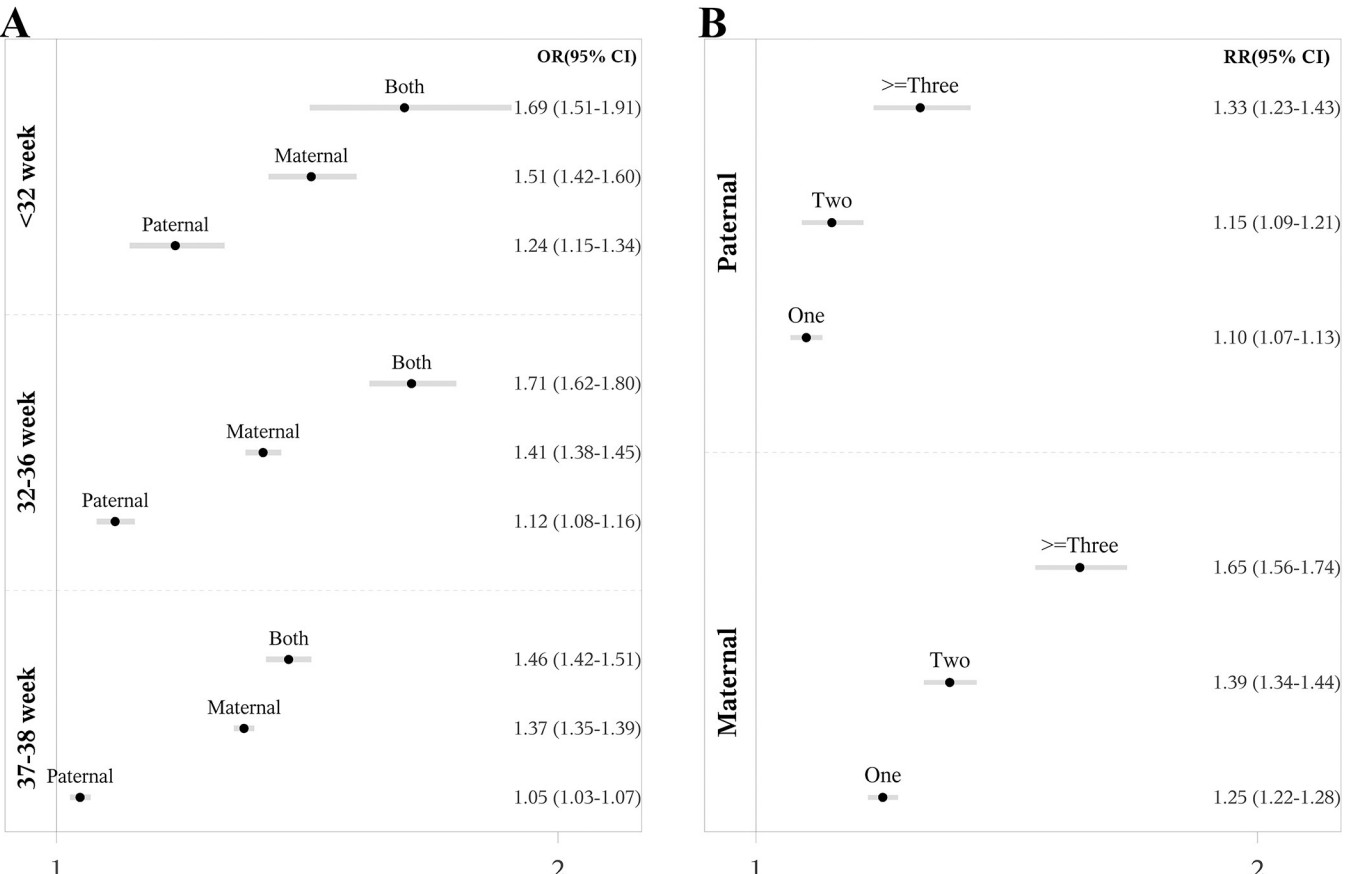

**Fig 3. Parental psychiatric history and risk of preterm birth, by gestational age categories and by co-occurring different psychiatric disorders. Part A: Parental psychiatric history and risk of very preterm birth, moderate/late preterm birth, and early term birth versus full term birth.** ORs of GA <32 weeks (very preterm birth), GA 32 to 36 weeks (moderate/late preterm birth), and GA 37–38 weeks (early term birth) with 95% CIs, versus GA ≥39 weeks (full term birth), were calculated using multi-nominal logistic regression models adjusted for birth year by cubic natural splines with 5 knots. The dots in the figure represent ORs and lines represent 95% CIs. The reference group was offspring of neither parent with a psychiatric diagnosis. **Part B: Number of co-occurring psychiatric disorders from different diagnostic categories in fathers and mothers and risk of preterm birth.** RRs of preterm birth with 95% CIs were calculated using log-binomial regression models, adjusted for birth year by cubic natural splines with 5 knots and any psychiatric diagnosis of the other parent. The dots in the figure represent RRs and lines represent 95% CIs. Diagnostic categories of psychiatric diagnoses: psychoactive substance use, schizophrenia and other non-mood psychotic disorder, mood disorders, neurotic/behavioral disorders, neurodevelopmental disorders, emotional and behavioral disorders of childhood origin and intellectual disability, and other psychiatric disorder (not otherwise specified). CI, confidence interval; GA, gestational age; OR, odds ratio; RR, relative risk.

stress-related disorders [12,13], eating disorders [15,16], neurodevelopmental disorders [18,19], substance use [20], schizophrenia [21], and OCD [22].

Our results show a consistent risk pattern across different psychiatric diagnoses and gestational weeks. The underlying pathophysiology behind the observed association is complex. The risk of preterm birth is more pronounced in spontaneous than in non-spontaneous delivery. Stress and inflammation/infection are main causes of spontaneous preterm and interact with genetic susceptibility [40]. Maternal stress is also regarded as an important link between maternal psychiatric disorders and preterm birth through endocrine, behavioral, and/or inflammation/immune-mediated mechanisms [5]. Suffering from a psychiatric condition, or having a partner with a psychiatric condition during pregnancy, may be challenging to the mother. Increased psychological stress has been linked to higher levels of stress hormones such as cortisol, which may induce premature uterus contractions [40,41]. Accordingly, the highest point estimates for risk of preterm birth is observed for paternal and maternal stress-

related disorders. In addition, 2 meta-analyses show that paternal depression is associated with maternal depression [42,43]. Studies have identified factors such as violence, substance abuse, and smoking as important mediators for poor obstetric outcomes, including preterm delivery, in women with severe mental illness [44–46]. It is possible that the same social and environmental disadvantages that predispose to preterm birth cluster in couples in which both have mental illness [47]. A recent meta-analysis demonstrates that unemployment and financial strain are important risk factors for paternal depression [43]. In line with this, we found that adjustment for parental education, income, and age at delivery decreased risks associated with paternal psychiatric history but risk estimates remained essentially unchanged for maternal psychiatric history. This may indicate the involvement of environmental/social factors in the pathway between parental psychiatric history and preterm births, particularly for paternal associations. Still, genetic causes are the main drivers of risk for psychiatric disorders, and psychiatric disorders are among the most heritable diseases [6]. There is evidence for genetic components and gene–environment interactions in spontaneous preterm birth [1,7], and it is not clear if these are also associated with psychiatric conditions. It is possible that, through shared genetic or environmental factors, parents who carry genetic risk for psychiatric disorders may also carry a higher liability for preterm delivery. In our study, the risk of preterm birth increased with number of different psychiatric disorders in a parent. This dose–response relationship may also indicate that higher disease load, either genetic load or as a measure of disease severity, is linked to magnitude of the risk of preterm delivery. Consequently, if genetic, the pattern with lower association in fathers than in mothers suggests a risk driven by interactions between genetic susceptibility and environmental exposures; the genetic risk may in turn be modified by socioeconomic risks and other gene–environmental interactions [1,7].

At least 15% of couples giving birth in Sweden had a history of psychiatric diagnosis(es), which implies a substantial potential for public health initiatives. Paternal involvement during pregnancy has been shown to moderate maternal stress, and the lack of such paternal support is associated with adverse perinatal outcomes [48]. However, such psychological support may be lacking in which both members of the couple have psychiatric conditions. Linking psychological stressors with increased risk of preterm birth, factors such as social and environmental disadvantages are likely to cluster and psychological support to counteract such stress is more likely to be adversely affected in families were both parents have mental conditions. This raises the concern about preventive efforts in antenatal care and opens possibilities for clinical intervention, e.g., whether additional social support to these families may be needed. Causally, if we speculate that inherited genetic risk as well as "biological" or "psychological" stress can carry over to other populations, the results are generalizable beyond Sweden. However, risk from downstream factors, e.g., obesity and smoking, as well as public health interventions may differ between countries and populations.

## Supporting information

**S1 Checklist. STROBE Statement—checklist of items that should be included in reports of observational studies.**
(DOC)

**S1 Appendix. Supplementary Tables and Figures. Table A.** ICD codes for psychiatric disorders. **Table B.** Percentiles of gestational age in days by parental psychiatric history. **Table C.** Parental psychiatric history and relative risk of preterm birth, adjusting for maternal smoking and BMI. **Table D.** Parental psychiatric history before delivery and relative risk of very preterm birth (<32 weeks), moderate/late preterm birth (32–36 weeks), and early term birth (37–38 weeks) vs. full term birth (≥39 weeks). **Table E.** Number of co-occurring psychiatric

disorders in fathers and mothers and relative risk of preterm birth. **Table F.** Parental psychiatric history and relative risk of preterm birth pairwise comparisons between exposure groups—father only, mother only, and both parents. **Table G.** Parental psychiatric history before delivery and relative risk of preterm birth, by spontaneous delivery and non-spontaneous delivery. **Table H.** Parental psychiatric history before delivery and relative risk of preterm birth, by offspring sex. **Table I.** Parental psychiatric history before delivery and relative risk of preterm birth in singletons. **Table J.** Parental psychiatric history before delivery and relative risk of preterm birth, restricting to psychiatric disorders first being diagnosed 2 years before conception. **Table K.** Parental psychiatric history before delivery and relative risk of preterm birth, requiring at least 2 diagnoses over 30 days. **Table L.** Parental psychiatric history before delivery and relative risk of preterm birth, restricting to births from 2005 onwards. **Fig A.** Flow diagram illustrating the identification of the study cohort. **Fig B.** Density curve of gestational age (GA) distributions by parental psychiatric history. **Fig C.** Cumulative distribution of years between birth of the child and last previous diagnosis in fathers and mothers.
(PDF)

**S1 Text. Analysis plan.**
(PDF)

## Author Contributions

**Conceptualization:** Weiyao Yin, Ulrika Åden, Kari Risnes, Martina Persson, Abraham Reichenberg, Eero Kajantie, Sven Sandin.

**Formal analysis:** Weiyao Yin.

**Funding acquisition:** Sven Sandin.

**Project administration:** Weiyao Yin, Sven Sandin.

**Resources:** Sven Sandin.

**Supervision:** Jonas F. Ludvigsson, Abraham Reichenberg, Eero Kajantie, Sven Sandin.

**Writing – original draft:** Weiyao Yin.

**Writing – review & editing:** Jonas F. Ludvigsson, Ulrika Åden, Kari Risnes, Martina Persson, Abraham Reichenberg, Michael E. Silverman, Eero Kajantie, Sven Sandin.

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
