## [Editor Report · Decision Letter 0]

20 Dec 2022

Dear Dr Yin, 

Thank you for submitting your manuscript entitled "Paternal and maternal psychiatric history, gestational age and preterm birth" for consideration by PLOS Medicine.

Your manuscript has now been evaluated by the PLOS Medicine editorial staff as well as by an academic editor with relevant expertise and I am writing to let you know that we would like to send your submission out for external peer review.

Please re-submit your manuscript within two working days, i.e. by Dec 22 2022 11:59PM.

Kind regards,

Philippa Dodd, MBBS MRCP PhD

PLOS Medicine

---

## [Decision Letter · Decision Letter 1]

3 Mar 2023

Dear Dr. Yin,

Thank you very much for submitting your manuscript "Paternal and maternal psychiatric history, gestational age and preterm birth" (PMEDICINE-D-22-03876R1) for consideration at PLOS Medicine. 

[LINK]

In light of these reviews, I am afraid that we will not be able to accept the manuscript for publication in the journal in its current form, but we would like to consider a revised version that addresses the reviewers' and editors' comments. Obviously we cannot make any decision about publication until we have seen the revised manuscript and your response, and we plan to seek re-review by one or more of the reviewers. 

We expect to receive your revised manuscript by Mar 24 2023 11:59PM. Please email us (plosmedicine@plos.org) if you have any questions or concerns.

We look forward to receiving your revised manuscript. 

Sincerely,

Philippa Dodd, MBBS MRCP PhD

PLOS Medicine

plosmedicine.org

GENERAL

Please respond to all editor and reviewer request detailed below, in full.

Please number the lines in continuous sequence starting at 1 at the top of page 3

Please check carefully throughout for minor grammatical errors “was” instead of “were”, for example

Please ensure that the study is reported according to the STROBE guideline and include the completed STROBE checklist as Supporting Information. Please add the following statement, or similar, to the Methods: "This study is reported as per the Strengthening the Reporting of Observational Studies in Epidemiology (STROBE) guideline (S1 Checklist)."

When completing the checklist, please use section and paragraph numbers, rather than page or line numbers, as these often change in the event of publication.

COMMENTS FROM THE ACADEMIC EDITOR

It seems clear to me that this is an interesting paper and that there are 3 positive reviews. Definitely good to send manuscript back to authors with a Major Revision decision.

My only additional comment might be to ask the authors to speak a little bit more about mechanism. How might father+mother psychiatric history increase risk over that of just one of them (for example). Authors speak a bit about gene-environment interaction as a mechanism, but it is brief and not entirely convincing. Without a careful consideration of mechanism, paper runs the risk of being a little descriptive.

DATA AVAILABILITY STATEMENT

PLOS Medicine requires that the de-identified data underlying the specific results in a published article be made available, without restrictions on access, in a public repository or as Supporting Information at the time of article publication, provided it is legal and ethical to do so. Please see the policy at 

http://journals.plos.org/plosmedicine/s/data-availability

and FAQs at 

http://journals.plos.org/plosmedicine/s/data-availability#loc-faqs-for-data-policy

For each data source used in your study: 

TITLE

Please revise your title according to PLOS Medicine's style. Your title must be nondeclarative and not a question. It should begin with main concept if possible. "Effect of" should be used only if causality can be inferred, i.e., for an RCT. Please place the study design ("A randomized controlled trial," "A retrospective study," "A modelling study," etc.) in the subtitle (ie, after a colon).

ABSTRACT

Abstract Methods and Findings:

Line 14 and 15: are these percentages correct? Please ensure all numerators and denominators used to define percentages are clearly defined for the reader.

Please ensure that all numbers presented in the abstract are present and identical to numbers presented in the main manuscript text.

Line 15: “…to undiagnosed parents…” as written this could suggest a missed diagnosis which we don’t think you mean. Suggest, “…parents unaffected by psychiatric illness.” or something similar

Line 18: suggest “…were estimated…”

PLOS Medicine requests that outcomes are quantified with p values as well as 95% CIs. When reporting p values please report as p<0.001 or where higher as p=0.002, for example.

Suggest reporting statistical information as follows, RR 1.12 (95%CI [1.08, 1.15] p<0. or p=)

Note the removal of the = symbol, the use of parentheses and commas instead of hyphens (as these can be confused with reporting of negative values) to separate upper and lower bounds.

Please include any important dependent variables that are adjusted for in the analyses.

In the last sentence of the Abstract Methods and Findings section, please describe the main limitation(s) of the study's methodology

Abstract Conclusions:

Line 7: suggest “The risk increases when fathers had a positive psychiatric history…” instead

AUTHOR SUMMARY

The author summary should include 2-3 single sentence bullet points under each question.

We encourage you to review existing published articles for examples on our website here https://journals.plos.org/plosmedicine/

INTRODUCTION

Line 1, page 5: please replace “Background with “Introduction”

Please address past research and explain the need for and potential importance of your study. 

Indicate whether your study is novel and how you determined that. 

If there has been a systematic review of the evidence related to your study (or you have conducted one), please refer to and reference that review and indicate whether it supports the need for your study.

METHODS and RESULTS

Page 6, line 21: “…ultrasound examination is routinely offered,” would it be worth stating that this is to determine gestational age, for clarity.

Page 9, line 1: please remove the semi-colon after the sub-heading

Lines 4-7: as above, percentages presented don’t seem to correlate with the numerators and denominators presented here. Please clearly define the numerators and denominators used to derive percentages. Apologies if I have misunderstood something but perhaps all the more reason to revise/clarify!

As above, PLOS Medicine requests that outcomes are quantified with p values as well as 95% CIs, including in tables and figures (please see below). When reporting p values please report as p<0.001 or where higher as p=0.002, for example.

TABLES

Please ensure that each figure caption affiliated to tables clearly describes their content without the need to refer to the text.

where 95% CIs are reported please also report p values. Please do not use asterisks to represent p values. Please report p as p<0.001 or where higher as p=0.002, for example. If not reporting p values, for the purpose of transparent data reporting, please clearly state the reasons why not.

Table 1: to improve accessibility to the reader, suggest removing the % symbols, except in column headers or row headers where you define (%) for the reader.

Please see reviewer #1 comments regarding reporting of income which we agree with

Maternal/paternal age at delivery please include (years) presumably?

Table 2: As above, please include p values with 95% CIs. Please ensure that upper and lower bounds of CIs are reported on the same row to improve reader accessibility.

We suggest separating upper and lower bounds by commas instead of hyphens as these can be confused with reporting of negative values.

FIGURES

Please ensure that each figure caption clearly describes the figure content for the reader without the need to refer to the text.

Figure 1: in the figure caption please indicate that your analyses are unadjusted. Please consider avoiding the use of green and/or red to make the figure more accessible to those with colour blindness.

Figure 2 and 3: in the figure caption please clearly define the meaning of the dots (RR) and lines (95% CI) for the reader. Suggest that the size of the lines Vs the dots be more proportionally presented to look more like lines rather than blocks

DISCUSSION

Line 17, page 10: “In this prospective nationwide birth cohort…” this is not a prospective study. Please revise. Suggest, “…observational cohort study…” or something similar

Line 9, page 11: “for the first time” please avoid assertions of primacy which can be risky. Suggest to the best of our knowledge, or similar.

Line 12, page 12: please remove the sub-heading “Conclusion”

Please remove details of funding, conflict of interest and data availability statements from the end of the discussion and include only in the manuscript submission form when you re-submit your manuscript. 

Please remove the patients and public involvement and transparency statements all together.

Please move your ethics statement to the methods section of the main manuscript.

REFERENCES

For in-text reference callouts citations should be placed in square brackets preceding punctuation, as follows “for example [1,3,9].” Please note the absence of spaces between citations.

In the bibliography, please ensure that up to but no more than 6 author names are listed followed by et al., in the event that more than 6 authors contribute to an individual study. Please see our website for other reference guidelines https://journals.plos.org/plosmedicine/s/submission-guidelines#loc-references

Comments from the reviewers:

Reviewer #1: Alex McConnachie, Statistical Review

Yin et al report on a large retrospective study of the association between a history of psychiatric diagnoses and preterm birth, using nationwide data from Sweden. This review considers the statistical aspects of the paper.

Generally, these are very good. The data sources and statistical methods are well described, and the results are presented quite clearly. My comments are fairly minor.

Conceptually, I would see this more as a case-control study, than a cohort study, since the population is defined by the outcome, rather than the initial condition. This is a study of all live births, not a study of all pregnancies.

Equally, I would describe this as a retrospective study, not prospective, since the study is being conducted after the data have been collected. A prospective cohort study would have identified all pregnancies as they happened, and followed them up to outcome. Quite rightly, the authors note the potential bias from not knowing about miscarriages.

At times, the authors stray into using causal language, such as in the conclusion that "Psychiatric diagnoses in both parents impact gestational age", which should be avoided.

I note that the exposure includes psychiatric diagnoses during pregnancy. Would there be any value in looking only at long-standing diagnoses as a sensitivity analysis? Or just at episodes during the pregnancy?

The statistical analysis section begins by talking about looking at the shift in distribution of gestational age at delivery, which is good, but this is only done graphically. No regression is attempted that looks at GA as a continuous outcome. The authors do go beyond a binary logistic regression to look at GA in 4 categories, but they analyse this with a multinomial regression model. Would it make more sense to use an ordinal regression approach? Would this get closer to looking at the shift in distribution? Saying that, the current categorisation only separates out different degrees of preterm birth - three quarters of the population remain in a single group, so an ordinal regression is not looking at a shift across the whole distribution, only a shift into the left-hand tail.

The authors choose to report unadjusted associations as primary results, with adjusted associations as secondary, but they do not make much of these secondary results. I have not reviewed every result in detail, but my impression is that the maternal associations are relatively unaffected by these adjustments, whilst the paternal associations, even if they remain statistically significant, are often noticeably reduced after adjustment.

In the results, it states that "4,597 (6.25%) of those born preterm had fathers with psychiatric history". Is this the wrong way round? I.e. 6.25% of those who had fathers with psychiatric history were born preterm? The same thing applies to the rest of that sentence.

The modelling treats parental psychiatric history as a 4-level categorical variable (neither, mother only, father only, both) but would it make sense to look at maternal and paternal psychiatric histories as independent predictors? Is there any evidence that the combination of both parents having a psychiatric history is any more (or less) than the sum of the two separate associations? The authors use "dose-response" to talk about the associations across the four categories used, but I do not think that is quite the correct term, since the exposure is not numeric. It makes more sense when talking of associations with the number of maternal and paternal diagnoses. It could work if describing the association with the number of parents (0, 1, or 2) with a diagnosis.

The discussion section starts by saying that there was a shift towards earlier gestational age associated with parental history of psychiatric diagnoses. I am not sure this is a fair interpretation. The regression analyses only look at degrees of preterm birth, not the whole distribution, so all you can say is that parental history of psychiatric diagnoses is associated with an increased risk of preterm birth. This could be due to a shift in the distribution, but if parental psychiatric history was associated with an increase in the variance of gestational age at birth, but not the mean, then the same pattern could result. Regression analyses of gestation age at birth as a continuous variable would let you say there was a shift, but that might add a lot of material to the paper. 

Table 1 reports income to 6 significant figures. Is this too much? Also, what are the units?

In the contents for the supplement, the title of eTable 2 is wrong. In eTable 2 itself, why only report the lower percentiles? Why not show 50th, 75th, 90th, and 95th as well? Would that give a more complete description of the distribution, and the shift associated with parental psychiatric history?

Reviewer #2: This study investigates an understudied but important area, the contribution of paternal factors to pregnancy outcomes and offspring health. It has the advantage of large numbers by using the National Patient Register in Sweden. Psychiatric health is a timely topic given the reported increase in disorders during the pandemic.

In the introduction the paper highlights a data gap as "the association between maternal psychiatric history and preterm delivery has not been extensively studied across a wide range of psychiatric disorders. Rather, previous studies have generally focused on mood and anxiety disorders." This statement is not entirely accurate, as reference 4 by Männistö et al found an increased birth associated with maternal bipolar disease and schizophrenia. The present analysis does expand on additional disorders, although I would defer to a psychiatrist about the categories. I think the main novel contribution of the current paper is the ability to study paternal psychiatric disorders. 

The same comment about the 2nd data gap in the introduction. The current paper uses the same plots as the Männistö paper (ref 4), so it's not truly a data gap. I think the introduction would benefit from highlighting the 3rd data gap which is the most novel.

Methods

The definition of spontaneous and non-spontaneous preterm birth needs to be added to the "Psychiatric diagnoses and preterm birth" section on p. 6. 

It's not clear to me why cubic splines were used for birth year with 5 degrees of freedom. Perhaps it would help to add a reference. I defer to a statistician.

Results

All of the covariates included in the models should be provided in Table 1 - for example, BMI, smoking, etc. Also, preterm birth is an outcome and outcomes are usually presented in a separate table, not Table 1.

This point is minor but only 1 decimal place is needed for percents, particularly the % preterm birth (p. 9. Lines 5-7). 

Suggest using an alternative terminology for the reference group instead of "Undiagnosed parents". Perhaps "Neither parent with psychiatric disorder" or something along those lines. Also, is it "psychiatric history" or current "psychiatric disorder"? "History" means something different. Need to be technically precise with the terminology.

Reviewer #3: This study examines associations between parental psychiatric diagnoses and length of gestation. The study makes a valuable contribution to the field, and the results eloquently illustrate dose-response relationships between psychiatric diagnoses of the parents individually and in combination in relation to shortened gestation. Most of my comments are focused on language for presenting the methods and findings.

In the main descriptive results in the abstract and manuscript, the denominators for the percentages are not clear. In addition, it is not immediately clear where the numerator 15,540 in the abstract comes from. Accordingly, please explicitly specify that 15,540 infants were born preterm from parents one or both of whom had a psychiatric history. In addition, please ensure that it is clear that the percentages in the sentences beginning at line 14 in the abstract (page 3) and line 5 of the results (page 9) refer to the PTB rates within exposure strata, not the size of the exposure strata as a proportion of the overall analytic sample. I recommend reporting the total number of preterm infants and the overall PTB rate (at least in the main manuscript) for context. Finally, to improve clarity I would change "No parents" to "Neither parent" in the column heading in Table 1.

Given the heterogeneity in the literature around different GA windows and designations of PTB severity, I would explicitly define early term birth in the abstract (as you do at line 14 of page 5).

Please standardize the formatting of reference numbers to ensure consistency (some but not all are currently in superscript).

Lines 5-6: "Paternal support during pregnancy has been associated..."

Page 7, line 11: gestational age densities - this terminology is unclear to me, and I'm not sure whether it's a standard term. Perhaps "gestational age distributions"?

Page 5, lines 9-10: relied on modest sample sizes with mixed results - I recommend adjusting this to "relied on modest sample sizes and yielded mixed results"

In the methods (abstract and main manuscript), please clarify the definition of "Nordic parents", as it's unclear whether this refers to ethnicity, residence, citizenship, or another definition. If this inclusion criterion is based on ethnicity, that would need to be justified.

Please describe the follow-up time during which exposures (psychiatric diagnoses) were measured, and whether this varied within the study population. Given that outpatient diagnoses were available beginning in approximately 2001, births at the end of the study period would presumably have more complete information on outpatient psychiatric diagnoses than those at the beginning of the study period. It would likely also be informative to report how long before delivery the most recent diagnosis tended to be in mothers and fathers who had a prenatal psychiatric diagnosis (e.g., knowing how often you had to look back further than 2 years to find a diagnosis would give a measure of how many diagnoses would be missed in patients with <2 years of prenatal data).

Last 2 sentences of page 6: I recommend rephrasing this to "We categorized gestational age into preterm (<37 weeks) 23 and term (≥37 weeks) birth, and then further into very preterm (<32 weeks), moderate to late preterm (32-36 weeks), early term (37-38 weeks) and full term (≥39 weeks).

Page 7, line 5 - change "was obtained" to "were obtained"

Page 7, line 8 - change to "...≥ 3 years postgraduate education, PhD education) were defined at delivery."

Page 8, line 11 - add "therefore" before "we did not include them..."

The authors report on statistical significance in several places where the point estimates are more important:

Page 9, line 18 - instead of reporting that associations remained statistically significant in adjusted analyses, please describe the changes in point estimates.

Similarly, page 10, lines 13-14: The estimated preterm risks remained when restricting to singleton births (eTable 9). Please describe any changes in the point estimates in the restricted model compared to the principal analysis.

Top of page 11: After adjustment ..., the risk remained statistically significant. Please describe if and how the point estimates changed in the adjusted models rather than focusing on the absence of a change in statistical significance.

Page 11, line 3: I recommend adding a hyphen in "sex-specific differences"

Page 11, sentence at lines 21-22. Please clarify that "both the parents" refers to psychiatric diagnoses for the parents.

Page 11, last paragraph: To improve clarity, I recommend setting off the text "in Sweden approximately 20% of all births" between dashes (—) instead of commas. 

Page 12, lines 6-7: I recommend adjusting the text to read "...suggested paternal depression as a risk factor for preterm birth (24), but not for paternal schizophrenia"

Page 12, liners 8-9: Adjust to read "Increased risk of preterm delivery in mothers with a psychiatric history has been reported..."

Page 13, line 4: Remove the comma after "Sweden" and add a comma after "psychiatric diagnosis(es)"

Page 13, line 8: Are the results generalizable beyond Sweden? Phrasing this as a question strikes me as inappropriately informal, so I would consider re-phrasing to summarize what we do or don't know about the generalizability of the study findings.

Finally, I recommend rephrasing the second sentence of the conclusion (page 13, lines 14-15), as it seems to read awkwardly beginning at "even higher".

[LINK]

---

## [Decision Letter · Decision Letter 2]

12 May 2023

Dear Dr. Yin,

Thank you very much for re-submitting your manuscript "Paternal and maternal psychiatric history and risk of preterm and early term birth: a nationwide register-based study" (PMEDICINE-D-22-03876R2) for review by PLOS Medicine.

I have discussed the paper with my colleagues and the academic editor and it was also seen again by 3 reviewers. I am pleased to say that provided the remaining editorial and production issues are dealt with we are planning to accept the paper for publication in the journal.

[LINK]

We look forward to receiving the revised manuscript by May 19 2023 11:59PM.   

Sincerely,

Philippa Dodd, MBBS MRCP PhD

PLOS Medicine

plosmedicine.org

Requests from Editors:

GENERAL

Thank you for your very detailed and considered responses to previous editor and reviewer comments. Please see below for further minor revisions which we require you address prior to publication.

TITLE

Please make reference to the study’s country of origin in the title, for example.’…a nationwide register-based study from Sweden’ or ‘…a nationwide study using Swedish registers’

ABSTRACT

Thank you for your clarifications around numerators, denominators & percentages reported here. We suggest for additional clarity detailing the number of preterm births in those unaffected by psychiatric illness before detailing those that are affected, for example (or similar):

‘Among the 1,488,920 infants born throughout the study period, 1,268,507 were born to parents without a psychiatric diagnosis and 73,094 (5.8%) were born preterm. 4,597 of 73,500 (6.3%) infants born to fathers with a psychiatric diagnosis were preterm. 8,917 of 122,611 (7.3%) infants were born preterm to mothers with a psychiatric diagnosis and 2,026 of 24,302 (8.3%) infants born to both parents with a psychiatric diagnosis were born preterm.’

Please also amend the same piece of data reporting in the results section of your manuscript (line 188 onwards).

AUTHOR SUMMARY

Thank you for including an author summary. Please include bullet points preceding each individual statement.

Line 44 – please make sentencing beginning ‘Earlier studies…’ a separate bullet point

Line 49 – suggest diagnosed ‘with’ instead of ‘of’

Line 50 suggest ‘(average relative risk [RR] =1.12)’ and thereafter ‘average RR…’ please note however, comments from reviewer #2 (please see below) regarding the presentation of average RR Vs the range of risk, which we agree with.

Line 52 – sentence beginning ‘For both…’ suggest making this a separate bullet point.

Line 57 – ‘Psychiatric diagnoses in both parents are associated with gestational age

INTRODUCTION

Line 64 – ‘…causes [1, 6, 7]…’ please remove all spaces between citations for in-text reference callouts to read as follows ‘…causes [1,6,7]…’. Please check and amend throughout all sections of the manuscript including supporting files where relevant. 

Line 66 – ‘systematical’ suggest ‘systematic’

RESULTS

As above, please makes amendments at line 188 to improve clarity.

STATISTICAL REPORTING

In the abstract you use commas to separate upper and lower bounds (which we favour) and in the main manuscript results section, hyphens separate upper and lower bounds. Suggest using commas instead to prevent confusion with reporting of negative values. See line 208 onwards.

DISCUSSION

Line 276 – ‘…gestational age range…’ instead perhaps

Line 283 – suggest ‘Only one study examined the risk of birth before full term. It included only mothers with psychiatric disorders but reported an increased risk of births before 39 weeks, consistent with our findings…’ or similar for improved clarity/grammar.

SOCIAL MEDIA

To help us extend the reach of your research, if not already done so, please detail any Twitter handles you wish to be included when we tweet this paper (including your own, your coauthors’, your institution, funder, or lab) in the manuscript submission form when you re-submit the manuscript.

Comments from Reviewers:

Reviewer #1: Alex McConnachie, Statistical Review

I thank the authors for their consideration of my original comments.

I found their responses to be entirely satisfactory. I feel the changes made are for the better, and their reasons for not making changes in response to some comments are fully justified. I have no further comments to make.

Reviewer #2: The authors have adequately responded to my comments. I have nothing further.

Reviewer #3: The has been improved substantially in line with the reviewers' comments. I have several minor substantive suggestions.

lines 50-52: It may be more informative to present the range of relative risks in each of the three exposure categories (fathers diagnosed, mothers diagnosed, both parents diagnosed) instead of or in addition to the average RR in each category.

75-77: evidence suggests that advancing birth, even by as little as 1 or 2 weeks before full term, can affect neonatal morbidity [26] and future health outcomes [2]. The phrase "advancing birth" suggests an intervention to change the length of gestation, which speaks to one of the main challenges regarding preterm birth and shortened gestation in general, i.e., PTB is associated with adverse outcomes later on, but it's not clear whether interventions to prolong gestation would also reduce these other sequelae (as both PTB and later adverse outcome may stem from the same underlying problem). Accordingly, I would rephrase this sentence to clearly avoid causal language, e.g.: "evidence suggests that even moderately shorter gestation is associated with neonatal morbidity ..."

Following are additional comments focused on language.

Line 175: before two years of conception - I recommend "prior to two years before conception"

Similarly, at line 238: "psychiatric diagnoses first diagnosed at least two years before conception"

176: to see how much - change to "To determine the extent to which the results were driven by individuals with provisional diagnoses"

89-90: I suggest "parents born in Sweden, Finland, Norway, Denmark or Iceland"

193 and elsewhere - "25th to 75th percentiles" can be changed to "interquartile range" (or "IQR")

197: was shown in Figure C in S1 Appendix - change to "is shown..."

13-14: We extended the analysis beyond the PTB - I suggest "... beyond PTB"

25-26: Please report which disorders the 3 different estimates correspond to (as you do in the main text, lines 217-219).

49-50: when fathers were diagnosed of different psychiatric disorders - I suggest "... diagnosed with different psychiatric disorders"

59: could have impact on gestational age - I suggest "could have an impact ..."

66: One recent systematical review - change to "... systematic review"

73-74: a rigorous study addressing the combined effect of psychiatric disorders in the couples is notably absent - I suggest "a rigorous study addressing the combined effect of psychiatric disorders in couples ..."

91: The fathers were identified - I suggest "Fathers were identified"

94: Psychiatric diagnoses from specialist care were extracted in the Swedish National Patient Register (NPR) - "... extracted from ..."

97: The data quality of the NPR has been verified [29] , and validated - remove comma before "and validated"

100-101: European Unions Horizon 2020 research and innovation program - change to "European Union's Horizon 2020 research and innovation program"

135-136: Where a linear regression is analysing differences in the mean between exposures - I suggest "Whereas a linear regression analyses differences in the mean between exposures"

139: the quantile regression can estimate at which gestational age - remove "the"

145-146: Then we examined the risk by six psychiatric diagnostic categories and by subtypes under the categories. I suggest "We then examined the risk by six psychiatric diagnostic categories and by subtypes within each category."

194 and 195: change "2 year" to "2 years"

205: typo for "gestational age"

244: spanning over 20 years, and including 1.5 million births - I would remove the comma before "and"

252-253: the risk was largely similar for maternal associations, but noticeably reduced for paternal associations, although remained statistically significant. I suggest "the risk was largely similar for maternal associations but was noticeably reduced for paternal associations, although it remained statistically significant."

257-278: gestational age determined using ultra sound - "ultrasound" 1 word (as at line 113)

258-259: a publicly financed health care system with equal access - I suggest "... with universal access"

264-265: Although we have examined the risk by diagnostic categories and subtypes of psychiatric disorders - I suggest "Although we examined ..."

302-303: Increased psychological stress has been linked to higher levels of stress hormones as cortisol which may induce premature uterus contractions - I suggest "... higher levels of stress hormones such as cortisol, which may induce premature uterine contractions"

318-319: it is not clear if these are associated also with psychiatric conditions - I suggest "it is not clear if these are also associated with psychiatric conditions"

330-331: However, such psychological support is probably lacking in which both of the couples have psychiatric conditions. Change "in which both of the couples" to "in which both members of the couple". I would also consider changing "is probably lacking" to "may be lacking" to highlight the somewhat speculative nature of the statement and avoid an accusatory tone.

335-336: whether additional social support to these families is need - I would change "is need" to "may be needed"

338-339: However, risk from downstream factors, e.g., obesity and smoking, as well as public health and intervention may differ between countries and populations. The phrase "public health and intervention" is unclear. Should this perhaps be something like "public health initiatives" or "public health interventions"?

[LINK]

---

## [Editor Report · Decision Letter 3]

1 Jun 2023

Dear Dr Yin, 

On behalf of my colleagues and the Academic Editor, Professor Mark Tomlinson, I am pleased to inform you that we have agreed to publish your manuscript "Paternal and maternal psychiatric history and risk of preterm and early term birth: a nationwide study using Swedish registers" (PMEDICINE-D-22-03876R3) in PLOS Medicine.

Prior to publication we require that you make the following revisions:

1) AUTHOR SUMMARY

Line 63 – suggest ‘These data suggest that the presence of psychiatric diagnoses in either one or both parents impacts gestational age at birth.’ Or similar

Line 64 – suggest ‘Whether additional social and psychiatric support and prenatal care to families with a positive psychiatric history could mitigate against this warrants further investigation.’

2) SUPPORTING FILES

S1 text – as part of your formatting changes (detailed below) please ensure you apply PLOS Medicine's referencing format to your statistical analysis plan.

PRESS

Best wishes,

Pippa 

Philippa Dodd, MBBS MRCP PhD 

PLOS Medicine